# *Drosophila* Models Reveal NAT Complex Roles in Heart Development and Enable Functional Validation of Congenital Heart Disease Variants

**DOI:** 10.3390/cells14201596

**Published:** 2025-10-14

**Authors:** Jun-Yi Zhu, Hannah Seah, Hangnoh Lee, Hanhan Liu, Zhe Han

**Affiliations:** 1Center for Precision Disease Modeling, Department of Medicine, University of Maryland School of Medicine, Baltimore, MD 21201, USA; 2Division of Endocrinology, Diabetes, and Nutrition, Department of Medicine, University of Maryland School of Medicine, Baltimore, MD 21201, USA

**Keywords:** N-terminal acetyltransferase, congenital heart disease, *Drosophila*, NAA16 variant validation, heart development

## Abstract

N-terminal acetylation, catalyzed by N-terminal acetyltransferase (NAT) complexes, is one of the most prevalent protein modifications in eukaryotic cells, yet its role in heart development remains poorly understood. Here, we use *Drosophila* as an in vivo platform to investigate the functions of NAT complex components in cardiac development and congenital heart disease (CHD). Focusing on the NatA complex, we showed that cardiac-specific knockdown of each of its three subunits (*Naa15-16*, *vnc*, and *san*) led to developmental lethality, structural disorganization, fibrosis, and impaired cardiac function in *Drosophila*. Remarkably, human NAA16 completely rescued the cardiac defects in *Naa15-16* silenced *Drosophila*, whereas a CHD-associated variant (NAA16-R70C) failed to do so, providing direct functional evidence of its pathogenicity. Together, these findings suggest the NatA complex as a critical regulator of heart development and provide functional validation linking variants in NatA complex genes to CHD. Further studies in mammalian models will be required to provide additional supporting evidence.

## 1. Introduction

Protein N-terminal acetylation is one of the most prevalent modifications in eukaryotic cells, occurring in nearly 80–90% of human proteins [1,2]. This modification is catalyzed by a group of enzymes known as N-terminal acetyltransferases (NATs), which transfer an acetyl group from acetyl-CoA to the N-terminal amino acid of a nascent protein [3,4]. NATs play essential roles in modulating protein stability, localization, interaction, and function [5,6,7]. In mammals, eight major NAT complexes (NatA–H) have been identified, each with distinct catalytic and auxiliary subunits and substrate specificities (Figure 1A) [8]. Among these, the NatA complex, which is composed primarily of the catalytic subunit NAA10 or NAA11 and the auxiliary subunit NAA15 or NAA16, is the most extensively studied and is responsible for acetylating a broad range of proteins following initiator methionine removal [9,10].

Although N-terminal acetylation has been widely recognized as a critical regulator of proteome function, its roles in organ-specific development and disease have only recently gained attention. In particular, emerging evidence points to the involvement of NAT complexes in cardiac development and disease. Genetic studies have implicated mutations in NAT components, especially those of the NatA complex, in congenital heart diseases (CHDs) [11,12,13]. For instance, loss-of-function mutations in NAA15 and NAA16 have been linked to syndromic forms of CHD, suggesting that protein acetylation may play an essential role in heart development [11,12,13]. Multiple studies suggested that dysregulation of NAT could cause various cardiac diseases, including cardiac arrhythmia and cardiomyopathy [14,15]. Reduction in NAA15 levels leads to altered expression of 562 proteins and changes in the N-terminal acetylation of 9 proteins, which are implicated in NAA15-mediated CHD [15]. Several case reports have linked mutations in NAT10 to Ogden syndrome, a condition in which cardiac defects are among the major clinical manifestations [16,17,18]. Mutations in NAA30 have also been reported to be potentially associated with heart disease [19]. However, no direct in vivo evidence has demonstrated that NAT complexes are required for heart development, and mechanistic studies linking NAT dysfunction to cardiac phenotypes remain limited. Furthermore, a comprehensive investigation to determine which specific NAT complexes are involved in heart development has not yet been conducted.

Recent genomic efforts, including those by the NIH Pediatric Cardiac Genomics Consortium (PCGC), have expanded the landscape of CHD-associated genes and revealed multiple de novo variants in NAT complex components, with NAA15 and NAA16 among the most frequently affected [11,12,13]. Although bioinformatic tools predict many of these candidate genes to be disease-associated, the lack of in vivo evidence presents a significant challenge in confirming their relevance to CHD. Additionally, the chance of the genetic variant identified in a given CHD gene causing the disease is relatively low [20,21]. A mutation identified in CHD patients may not affect the protein structure and function, which will not cause CHD. A large number of genetic variants identified in CHD patients remain unconfirmed regarding their association with the disease. This gap is largely attributed to the high cost and long timelines associated with mammalian models commonly used for such studies. Consequently, there is an urgent need for a low-cost, high-throughput animal model to validate the disease association of these candidate genes and their genetic variants.

*Drosophila* melanogaster has emerged as a powerful model for studying heart development due to its conserved cardiac structure, rapid genetic manipulation, and suitability for high-throughput in vivo functional assays [22,23,24]. The *Drosophila* heart, though structurally simpler, shares significant molecular and functional similarities with the vertebrate heart, particularly during early developmental stages [25,26]. Additionally, *Drosophila* provides a tractable platform for validating disease-associated variants through tissue-specific genetic manipulations and phenotype-rescue strategies [27].

In this study, we investigate the roles of NAT complexes, particularly the NatA complex, in heart development using *Drosophila* models. The NAT complex components are highly conserved between human and *Drosophila*, indicating their conserved roles in heart development during different species (Figure 1B). We first characterize the expression and function of several NAT complex components in the developing *Drosophila* heart. We then apply in vivo genetic approaches to dissect the functional consequences of cardiac-specific knockdown of NAT components, with a focus on *Naa15-16*, the *Drosophila* homolog of human NAA15 and NAA16. Furthermore, we employ gene replacement strategies that we developed previously [27,28,29] to validate a missense variant of NAA16 (R70C) identified in CHD patients, providing critical insights into its pathogenicity. Our findings reveal essential roles for NatA complex components in heart development and demonstrate the utility of *Drosophila* as an efficient platform for functional validation of CHD-associated genetic variants.

## 2. Results

### 2.1. Silencing NatA Complex Components Naa15-16, Vnc, or San Impaired Drosophila Survival

First, we examined the expression patterns of genes encoding the NAT complexes in the *Drosophila* embryonic heart at different developmental stages. We previously published a single-cell RNA-seq dataset that includes the gene expression levels of all genes encoding NAT complex components [30]. We reanalyzed single-cell RNA-seq data from cardiac cells collected across key embryonic stages, spanning the migration of bilateral cardiac progenitors (stage 13 onward) to the formation of the linear heart tube [30]. Among these genes, NatA complex component *Naa15-16* displayed consistently high and stable expression in cardiac progenitor cells across all developmental stages (Appendix A). Other NAT complex components, including *san*, *Naa20A*, and *Naa30A*, were also expressed in cardiac progenitor cells, but at comparatively lower levels (Appendix A). Notably, *vnc* expression was undetectable in cardiac progenitor cells at any of the examined stages, suggesting it was not among the highly expressed genes captured in this single-cell RNA-seq dataset.

Next, we used the *Drosophila* UAS-Gal4 system [31] to specifically knock out individual components of the NAT complexes in the heart. This was achieved by crossing the heart-specific Gal4 driver, 4X*Hand*-Gal4 [27], with available UAS RNAi lines NAT complexes component obtained from the Bloomington Drosophila Stock Center. The UAS RNAi lines obtained from the Bloomington Drosophila Stock Center can achieve 60–70% reduction using 4X*Hand*-Gal4 compared to controls (Appendix A). Cardiac-specific silencing of NatA complex components *Naa15-16*, *vnc*, and *san* significantly reduced eclosion rates, as indicated by the decreased number of adult flies successfully emerging from pupal cases (Figure 2A). Furthermore, the few adult flies that did eclose exhibited markedly shortened lifespans compared with controls (Figure 2B). These results the essential roles of the NatA complex components, Naa15-16, vnc, and san, in maintaining normal heart development and function in *Drosophila*. We did not observe any significant changes when silencing other NAT complex components in the *Drosophila* heart (Figure 2A,B).

### 2.2. Silencing NatA Complex Components Naa15-16, Vnc, or San Induced Cardiac Structural and Functional Defects in Adult Drosophila

Cardiac-specific knockdown of the NatA complex components *Naa15-16*, *vnc*, or *san* caused early lethality and markedly reduced lifespan (Figure 2A,B). To investigate the underlying cardiac abnormalities, we examined heart morphology in these flies. Notably, silencing Naa15-16 resulted in a complete loss of the heart structure in adult *Drosophila* (Figure 2C). We did not detect any cardiac myofibrillar organization and cardiac fibrillar density (Figure 2D). Further work is needed to determine how heart structures are lost in Naa15-16–silenced *Drosophila*. Knockdown of *vnc* or *san* caused severe disorganization of cardiac actin filaments, as revealed by phalloidin staining (Figure 2C), and significantly reduced cardiac muscle fiber density (Figure 2C,D). In addition, we observed marked accumulation of Pericardin in hearts with *vnc* or *san* knockdown (Figure 2C,E). Notably, this collagen signals detected by the pericardin antibody were completely absent in the hearts of *NAA15-16*-IR *Drosophila* (Figure 2E). This excessive Pericardin deposition observed upon *vnc* or *san* knockdown indicates fibrotic remodeling, a hallmark of cardiac injury [27,32,33].

To access cardiac functional defects resulting from *vnc* or *san* silencing, we employed optical coherence tomography (OCT), which serves as the *Drosophila* equivalent of echocardiography in humans [34,35]. The orthogonal view provided by OCT enables real-time measurements of heart tube diameter and heart period (Figure 3A). Because heart-specific silence *Naa15-16* led to a complete loss of the heart structure in adult *Drosophila*, no cardiac signals could be detected using OCT. We observed that silencing *vnc* or *san* significantly increased the heart period compared to control flies (Figure 3B), but did not alter the systolic or diastolic diameters (Figure 3C–E). In contrast, silencing other NAT complex components did not produce any notable morphological or functional heart defects (Figure 2 and Figure 3).

Altogether, these findings demonstrated that components of the NatA complex are essential for maintaining normal cardiac structure and function. Among them, *Naa15-16* stands out due to its consistently high expression in the developing *Drosophila* heart and the severe phenotypes observed upon its knockdown, highlighting its particularly critical role within the NAT complexes. However, further work is needed to perform detailed analyses of heart changes due to NatA complex components silencing during embryogenesis and larval developmental stages.

### 2.3. Expression of Wild-Type Human NAA16 and Its CHD-Associated Variant in the Drosophila Heart Did Not Cause Any Detectable Structural and Functional Cardiac Defects

Recent studies from the Pediatric Cardiac Genomic Consortium (PCGC) have identified more than 1000 candidate genes potentially associated with CHD [11,12,13]. Among them, several *de novo* mutations have been found in genes encoding components of the NAT complexes (Table 1). Notably, NAA15 and NAA16, which share the same *Drosophila* homolog, *Naa15-16*, are recurrently affected. Most variants in these genes are frameshift mutations, predicted to cause loss of function and considered pathogenic (Table 1). However, one missense variant R70C in NAA16 remains of uncertain significance regarding its disease association (Table 1). The detailed information of this R70C variant was previously reported (RadialSVM score: 0.314, RadialSVM pred: D, ExAC Overall: 2.49E-05, HHE Rank: 50.7, pLI score: 0.00) [11]. We compared the amino acid sequences between *Drosophila* and humans and confirmed that the mutant residue R70 is conserved in both species. This residue is located within the tetratricopeptide repeat (TPR) domain, which is also conserved between *Drosophila* and humans. TPR domain can act as modular docking sites, enabling proteins to form multi-protein complexes [36].

Since the missense variant R70C in NAA16 was identified in a heterozygous state in a CHD patient, we first sought to determine whether it exerted a dominant-negative or gain-of-function effect. To this end, we performed cardiac-specific overexpression of either wild-type human *NAA16* or the R70C variant in *Drosophila* hearts. We found cardiac-specific overexpression of the R70C variant did not produce additional phenotype compared with wild-type NAA16 (Figure 4). Further, neither construct induced morphological or functional cardiac abnormalities (Figure 4), indicating that the R70C variant may not function in a gain-of-function or dominant-negative mutation.

### 2.4. Cardiac Structural and Functional Defects Caused by Naa15-16 Silencing in Drosophila Can Be Rescued by Expression of Wild-Type but Not Mutant Human NAA16

To further validate the function of the NAA16 variant identified in the CHD patients, we performed rescue experiments in *Drosophila* using the gene replacement strategy we developed previously [27,28,29,38,39] (Figure 5A). First, we assessed the functional conservation between *Drosophila Naa15-16* and human *NAA16* to establish a platform for variant validation. Cardiac-specific expression of wild-type human *NAA16* in the *Naa15-16* silenced *Drosophila* completely rescued the morphological and functional cardiac defects, confirming their conserved function (Figure 5). In contrast, the NAA16 R70C variant failed to restore the normal cardiac morphology (Figure 5B). Because heart-specific silencing of *NAA15-16* resulted in a complete loss of cardiac structure in adult *Drosophila*, no cardiac signals could be detected by OCT in *Drosophila* with *NAA15-16* silencing or those together with expressing the NAA16 R70C variant. These results suggest that the R70C variant functions as a hypomorphic allele, exhibiting partial loss of activity that is insufficient to sustain normal cardiac development in the absence of endogenous *Naa15-16*.

## 3. Discussion

### 3.1. Drosophila as a High-Throughput Platform for Identifying Genes Required for Heart Development

*Drosophila* offers a powerful, high-throughput genetic system for dissecting conserved molecular mechanisms of heart development [22,23,24]. Though late cardiogenesis anomalies in humans involve chambers or structures that do not have direct morphological counterparts in *Drosophila*, *Drosophila* is considered a valuable model for studying early cardiogenesis defects because many signaling pathways, transcriptional regulators, and structural proteins involved in vertebrate heart development are highly conserved in flies [40,41,42]. Perturbations in these conserved components often produce analogous phenotypes, including defects in myocardial organization, contractility, fibrosis, and arrhythmia [23,32,43].

Importantly, *Drosophila* allows rapid in vivo testing of gene function and variant pathogenicity in the context of a beating heart [44,45,46]. Although the *Drosophila* heart does not recapitulate chamber-specific features, it provides a genetically tractable and high-throughput system to uncover conserved pathways and validate CHD-associated variants before moving into more complex mammalian models [47,48,49]. Its simple tubular heart shares evolutionary and functional conservation with the vertebrate heart, and genetic tools such as tissue-specific RNAi and CRISPR enable rapid gene functional testing. Our previous large-scale screens have successfully uncovered essential cardiac genes conserved in humans, demonstrating the utility of *Drosophila* as a platform for functional annotation of CHD candidate genes [27]. Here, we build upon this framework by leveraging the *Drosophila* heart to interrogate the role of NAT complex components in cardiac development, which has never been accessed in mammals.

### 3.2. The NatA Complex Is Required for Heart Development and Function

N-terminal acetylation, mediated by NAT complexes, is one of the most prevalent protein modifications in eukaryotes, regulating protein stability, localization, and interactions [5,6,7]. Among these, the NatA complex, which is composed of NAA10 or NAA11 (catalytic) and NAA15 or NAA16 and NAA50 (auxiliary subunit) (Figure 1A), is responsible for modifying nearly 40–50% of the proteome. In our study, we focused on the NatA complex components *Naa15-16*, *Drosophila* homolog of *NAA15* and *NAA16*, *vnc*, *Drosophila* homolog of *NAA10* and *NAA11*, and *san*, *Drosophila* homolog of *NAA50.* They were all highly conserved from human to *Drosophila* and essential for *Drosophila* heart development. Cardiac-specific knockdown of these genes resulted in structural heart defects, early lethality, and reduced lifespan.

The observed loss of myofibrillar organization and abnormal extracellular matrix deposition following *Naa15-16*, *vnc*, or *san* silencing point toward roles in cytoskeletal integrity and ECM remodeling. Given that N-terminal acetylation regulates protein stability and interactions [2,7,50,51], it is possible that NatA substrates include sarcomeric proteins, actin-binding factors, or ECM regulators essential for cardiac morphogenesis. Future studies using proteomic profiling and biochemical assays will help define these substrates and clarify how loss of NatA activity disrupts cardiac structure and function.

Notably, data from the PCGC revealed that approximately 20% of CHD-associated variants are linked to both CHD and neurodevelopmental disabilities (NDD), while only about 2% are associated with isolated CHD [12]. The patients harboring the NAA15 variants presented with CHD but exhibited no signs of NDD [12], suggesting that NAA15 plays a heart-specific role in development, rather than contributing to broader neurodevelopmental phenotypes.

These results provide strong in vivo evidence supporting a critical and previously underappreciated role for NatA complex components in heart development and maintenance. No studies have investigated the roles of *Naa15-16*, *vnc* and *san* in *Drosophila* heart development. Their exact functions at different stages of cardiac development remain unknown. Further studies were necessary to examine these NatA complex components to define their role in heart development across developmental stages, as well as their downstream targets and molecular pathways.

### 3.3. Drosophila Enables Variant-Level Validation

One of the major challenges in CHD research is assessing the functional impact of the growing number of candidate variants identified through large-scale exome and genome sequencing efforts. While bioinformatic tools can provide preliminary predictions, experimental validation is essential to distinguish pathogenic variants from benign polymorphisms. However, traditional mammalian models such as mice are time-consuming, costly, and low throughput, making them suboptimal for systematic variant testing. In contrast, *Drosophila* offers a genetically tractable, scalable, and cost-effective in vivo system that enables efficient evaluation of both gene function and variant pathogenicity in the context of a functioning heart.

In this study, we used the *Drosophila* heart model to perform variant-level validation of a missense mutation, R70C, in the human *NAA16* gene, which was identified in CHD patients by the PCGC. Using a “gene replacement” strategy, we first established that wild-type human *NAA16* could fully rescue the severe cardiac morphological and functional defects caused by *Naa15-16* knockdown in the fly heart, thereby demonstrating functional conservation between human and *Drosophila* orthologs. We then introduced the disease-associated NAA16-R70C variant in the same rescue model. This mutant form failed to restore heart structure or function, indicating an LOF effect and confirming the variant pathogenicity in vivo. Additionally, the R70C variant did not alter heart development when overexpressed, but failed to rescue the phenotype in the *Naa15-16*-IR model. This suggests that the variant protein is not gain-of-function or dominant-negative but rather represents a loss-of-function allele, retaining insufficient activity to support normal cardiac development in the absence of endogenous *Naa15-16*.

This approach provides compelling evidence for the utility of *Drosophila* in validating disease-associated alleles at the variant level. It demonstrates that specific human missense mutations can be functionally tested in a whole-organism context that recapitulates key aspects of cardiac development and physiology. Moreover, this strategy enables rapid screening of multiple variants to prioritize those most likely to contribute to disease phenotypes. As such, *Drosophila* represents a powerful complementary system to mammalian models for bridging the gap between genomic discovery and functional interpretation in the era of precision medicine.

### 3.4. Potentials of Drosophila as a Preclinical Model for CHD-Linked Ogden Syndrome and Therapeutic Testing

Ogden syndrome is a rare, X-linked developmental disorder caused by mutations in NAA10, which encodes the catalytic subunit of the NatA complex. Patients with Ogden syndrome exhibit a broad spectrum of clinical manifestations, including severe developmental delay, craniofacial dysmorphisms, hypotonia, and notably, CHDs [16,52,53]. Despite its clinical severity, the molecular mechanisms linking NAA10 dysfunction to cardiac phenotypes remain poorly understood, and no targeted therapies are currently available.

In this study, we showed that Ogden syndrome can be modeled in *Drosophila* by silencing *vnc*, the *Drosophila* homolog of *NAA10,* specifically in the heart. Loss of *vnc* in the *Drosophila* heart led to marked cardiac morphological abnormalities, including disorganization of myocardial actin filaments, reduced muscle fiber density, and excessive Pericardin accumulation, a hallmark of cardiac fibrosis. These phenotypes phenocopy aspects of human cardiac pathology and underscore the importance of NatA-mediated acetylation in maintaining cardiac structure and function.

Our findings provide in vivo support for the pathogenicity of NAA10 dysfunction in cardiac tissue and highlight *Drosophila* as a powerful model system for studying Ogden syndrome. The mechanistic role of NAA10 in heart development and its involvement in associated diseases requires further investigation. Importantly, *Drosophila* has previously been recognized as a high-throughput, genetically accessible model for validating candidate heart disease genes [27], as well as for exploring the molecular mechanisms and drug discovery pathways relevant to cardiovascular and other systemic diseases [54,55,56,57]. Future studies can leverage the *Drosophila* model to conduct small-molecule screens aimed at rescuing *vnc*-associated cardiac defects and identify the potential treatment for Ogden syndrome. Once promising drugs are identified in the *Drosophila* model with *vnc* silencing, further validation in mammalian models will be required.

Taken together, our findings demonstrate that the NatA complex plays a conserved and essential role in heart development. Using *Drosophila*, we developed a high-throughput platform for validating CHD-associated genes and variants and identified a path forward for modeling and potentially treating heart defects associated with Ogden syndrome. This work not only advances our understanding of NatA complex biology but also positions the *Drosophila* heart as a potentially beneficial preclinical tool for CHD research.

## 4. Materials and Methods

### 4.1. Drosophila Lines

*Drosophila* stocks were obtained from the Bloomington Drosophila Stock Center (BDSC; Indiana University Bloomington, Bloomington, IN, USA). The following lines were used in this study: UAS-*Naa15-16*-RNAi (BDSC ID 34990), UAS-*vnc*-RNAi (BDSC ID 56861), UAS-*san*-RNAi (BDSC ID 36632), UAS-*Sbat*-RNAi (BDSC ID 55161), UAS-*Naa20A*-RNAi (BDSC ID 36899), UAS-*Naa30A*-RNAi (BDSC ID 63983), UAS-*Naa35*-RNAi (BDSC ID 61901), Control w1118 (BDSCID 3605) flies were used in the crosses. The 4XHand-Gal4/Cyo driver was generated in our lab and used to direct cardiac-specific expression of RNAi-based silencing (-IR) constructs. The wild-type and R70C NAA16 cDNA were synthesized by GenScript (Piscataway, NJ, USA) and individually cloned into the pUAST-attB vector. Transgenic lines were generated via germline transformation into a fixed chromosomal docking site to ensure equivalent expression levels of the wild-type and variant NAA16 alleles in the *Drosophila* hearts.

### 4.2. Fly Quantitative RT-PCR Analysis

RNA was isolated from a single 4-day-old adult female fly using TRIzol reagent (Invitrogen, Waltham, MA, USA). RNA purity and concentration were assessed with a NanoDrop-1000 spectrophotometer (Thermo Scientific, Waltham, MA, USA). Total RNA (1 μg) was reverse transcribed using SuperScript IV (Invitrogen, Waltham, MA, USA). SYBR Green–based quantitative real-time PCR (Power SYBR Master Mix; Applied Biosystems, Waltham, MA, USA) was performed on a StepOnePlus system (Applied Biosystems) with gene-specific primer pairs (Integrated DNA Technologies, Coralville, IA, USA). Primer sequences are listed in Appendix A. Relative gene expression was calculated using the 2-DDCT method and normalized to Gapdh as an endogenous control. Four biological samples were analyzed per genotype.

### 4.3. Lethality at Eclosion

Eclosion lethality was assessed as the percentage of flies expressing an RNAi-based silencing construct (straight wings) that fail to emerge as adults, relative to their non-expressing siblings (curly wings) as previously described [58,59]. The eclosion lethal rate (percentage) was calculated as [((curly—straight)/curly) × 100]. At least 400 flies (female and male) were analyzed per genotype.

### 4.4. Adult Drosophila Survival Assay

*Drosophila* larvae were maintained at 25 °C to induce UAS-transgene expression. After eclosion, adult male flies were collected and kept at 25 °C, with 20 flies per vial. Mortality was recorded every 48 h as previously described [58,59]. One hundred flies were analyzed per genotype.

### 4.5. Immunochemistry

Adult *Drosophila* females (4 days old) were dissected and fixed for 10 min in 4% paraformaldehyde prepared in phosphate-buffered saline (1X PBS) as previously described [58,59]. Samples were incubated overnight at 4 °C with primary antibodies diluted in 2% bovine serum albumin (BSA; Sigma, St. Louis, MO, USA) with 0.1% Triton-X (Sigma, St. Louis, MO, USA) in 1X PBS. Following washes, secondary antibodies were applied for 2 h at room temperature in 2% BSA (Sigma, St. Louis, MO, USA) with 0.1% Triton-X (Sigma, St. Louis, MO, USA) in 1X PBS. Alexa Fluor 488 phalloidin (Thermo Fisher, Waltham, MA, USA; A12379) was used at a 1:1000 dilution. Mouse anti-Pericardin antibody (EC11; Developmental Studies Hybridoma Bank, Iowa City, IA, USA) was used at a 1:500 dilution, followed by Alexa Fluor 568 secondary antibody (Thermo Fisher, Waltham, MA, USA; A11011) at 1:1000 dilution. Confocal images were performed using a ZEISS LSM900 microscope with a 63X Plan-Apochromat 1.4 N.A. oil objective (ZEISS, Jena, Germany) and ZEN blue edition (version 3.0) acquisition software. Segment A3 of the heart was imaged by collecting Z-stacks. Control groups were imaged first to establish the laser intensity and exposure settings, which were then kept constant for all genotypes. The exposure time was adjusted to achieve approximately 70% of maximum signal saturation to allow consistent fluorescence intensity comparisons. Image processing was conducted in ImageJ (version 1.52a. 10 adult flies were imaged per genotype, and representative images are shown in the figures.

### 4.6. Heart Structural Analysis and Quantitation

*Drosophila* heart cardiac myofibril density and Pericardin deposition were quantified as previously described [58,59]. For each genotype, 10 adult flies (4-day-old females) were analyzed. Image processing and quantification were performed using ImageJ (version 1.52a). The Z-stack projections were examined, and image planes containing cardiac myofibrils were selected for analysis, avoiding the underlying ventral muscle layer. The cardiac myofibril number was quantified by using the MyofibrilJ plugin for Fiji (version 1.53q) [60]. The entire heart region within segment A3 was outlined using the freehand selection function in Fiji, and the number of cardiac myofibrils was measured. Cardiac myofibrillar density was calculated as the number of cardiac myofibril divided by the area of the selected heart region. Pericardin deposition was quantified in the same heart segment A3 based on the mean fluorescence intensity. Both cardiacmyofibril density and Pericardin deposition were normalized to those obtained from the control flies.

### 4.7. Optical Coherence Tomography (OCT)

Cardiac function in adult *Drosophila* was assessed using OCT [58,59]. The OCT system (Bioptigen) was constructed as described by the Biophotonics Group, Duke University, NC, USA. Further, 4-day-old adult female flies were anesthetized with carbon dioxide (CO_2_) for 3–5 min, then gently positioned on a glass plate with petroleum jelly (Vaseline) to immobilize them with the dorsal side facing the OCT imaging source. Flies were allowed to recover for at least 10 min prior to imaging to ensure full mobility and normal heart activity. For each genotype, ten control and ten RNAi-expressing flies were examined. OCT recordings were acquired from the same anatomical location, the cardiac chamber within abdominal segment A3 for all samples. Each measurement was obtained from three different positions within the abdominal segment A3, and the values were averaged to determine the cardiac diameter. M-mode imaging was used to record the heart wall movement during the cardiac cycle. ImageJ software (version 1.52a) was used to process the images. The diastolic dimension and systolic diameter were processed, measured, and determined based on three consecutive heartbeats. The heart period was determined by counting the total number of beats that occurred during a 15 s recording and then dividing 15 by the number of beats.

### 4.8. Statistical Analysis

Statistical tests were performed using PAST.exe software (latest v.4.13) [Natural History Museum, University of Oslo (UiO), Oslo, Norway]. Data normality was assessed using the Shapiro–Wilk test (α = 0.05). Normally distributed data were analyzed by a one-way analysis of variance (ANOVA) followed by a Tukey–Kramer post-test for comparing multiple groups. Non-normal distributed data were analyzed by a Kruskal–Wallis H-test followed by a Dunn’s test for comparisons between multiple groups. Values are presented as mean ± standard deviation (s.d.) Statistical significance was defined as * *p* < 0.05, ** *p* < 0.01, *** *p* < 0.001. Details for sample sizes used for quantitation have been provided in the figure legends.

## Figures and Tables

**Figure 1 cells-14-01596-f001:**
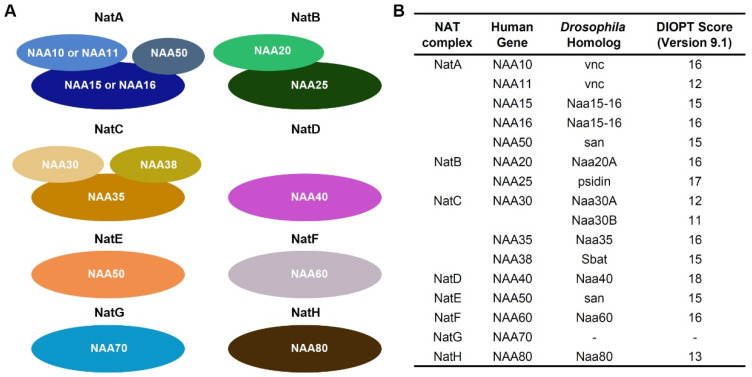
NAT complexes and their components. (**A**) Schematic representation of eight different NAT complexes. NAA10, NAA11, NAA20, NAA30, NAA40, NAA50, NAA60, NAA70, and NAA80 are the catalytic enzymes in each complex. (**B**) Human and *Drosophila* NAT complexes and their subunits. DIOPT, DRSC integrative ortholog prediction tool (version 9.1; max score Drosophila-human is 19).

**Figure 2 cells-14-01596-f002:**
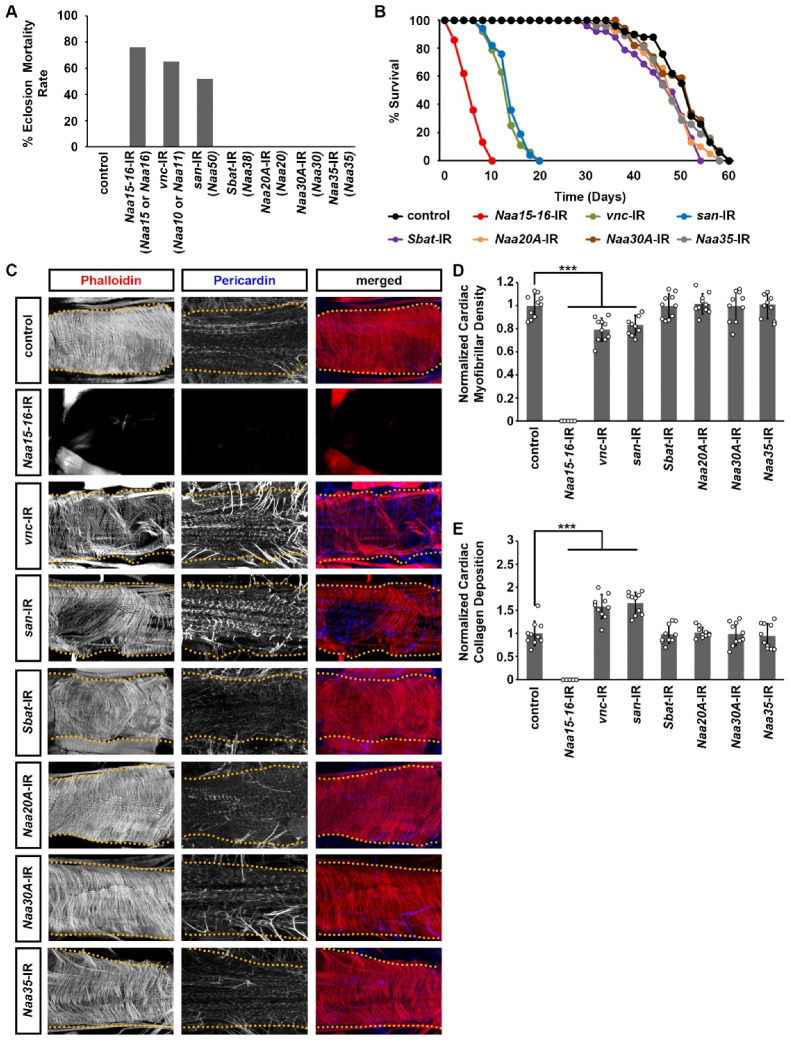
*Drosophila* survival and cardiac structure following heart-specific silencing of NAT complex components. (**A**) Eclosion lethality induced by 4X*Hand*-Gal4-driven expression of UAS-RNAi transgenes targeting individual NAT complex components. Crossed with a CyO (curly wing) balancer allowed identification of progeny: adults with curly wings (CyO) lacked transgene expression, whereas straight-winged flies expressed 4X*Hand*-Gal4 > RNAi. The eclosion lethal rate (percent) was calculated as [((curly–straight)/curly) × 100]. All lines also carried 4X*Hand*-Gal4. *n* = 400+ flies (female and male) per genotype. (**B**) Survival curves of adult flies expressing RNAi (-IR) transgenes targeting individual NAT complex components in heart cells. Control, 4XHand-Gal4+/−. *n* = 100 male flies (20 flies/vial) per genotype. (**C**) Representative confocal images of adult (4-day-old females) cardiac structure following the expression of UAS-RNAi transgenes targeting each NAT complex component (4XHand-Gal4). Cardiac actin myofibers were visualized by phalloidin staining (red). Pericardin was detected by immunofluorescence (blue). Dotted lines indicate heart tube boundaries. Control, 4XHand-Gal4+/−. (**D**) Quantitation of adult heart cardiac myofibrillar density relative to control (see image in (**C**)) [mean ± s.d.; *n* = 10 flies (4-day-old females) per genotype; Kruskal–Wallis H-test followed by a Dunn’s test; statistical significance: *** *p* < 0.001]. (**E**) Quantitation of adult heart cardiac collagen (Pericardin) deposition relative to control (see images in (**C**)) [mean ± s.d.; *n* = 10 flies (4-day-old females) per genotype; Kruskal–Wallis H-test followed by a Dunn’s test; statistical significance: *** *p* < 0.001].

**Figure 3 cells-14-01596-f003:**
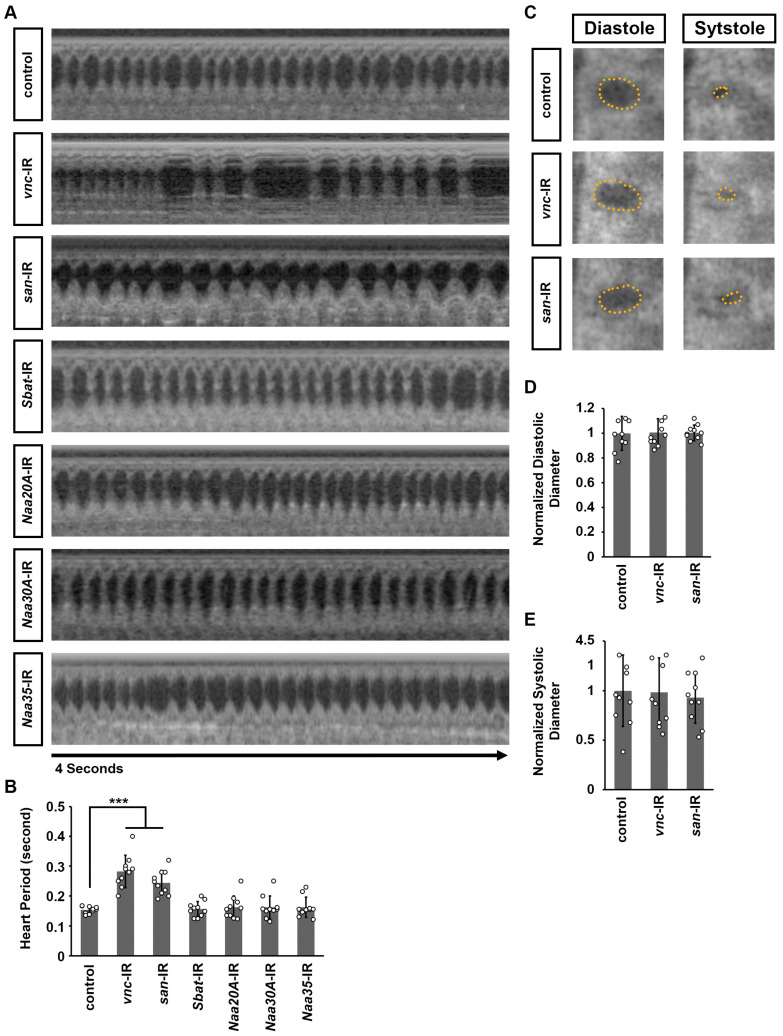
Cardiac function in flies following heart-specific silencing of NAT complex components. (**A**) Representative OCT images from heartbeat recording of *Drosophila* (4-day-old females) expressing UAS-RNAi transgenes targeting individual NAT complex components. Control, 4XHand-Gal4+/−. (**B**) Quantitation of heart period (see images in (**A**)). (**C**) Representative images for diastolic and systolic heart diameter are shown for flies that express UAS-RNAi transgenes targeting *san* and *vnc*. Control, 4XHand-Gal4+/−. (**D**) Quantitation of adult heart diastolic diameter (see images in (**C**)). (**E**) Quantitation of adult heart systolic diameter (see images in (**C**)). [mean ± s.d.; *n* = 10 flies (4-day-old females) per genotype; Kruskal–Wallis H-test followed by a Dunn’s test; statistical significance: *** *p* < 0.001].

**Figure 4 cells-14-01596-f004:**
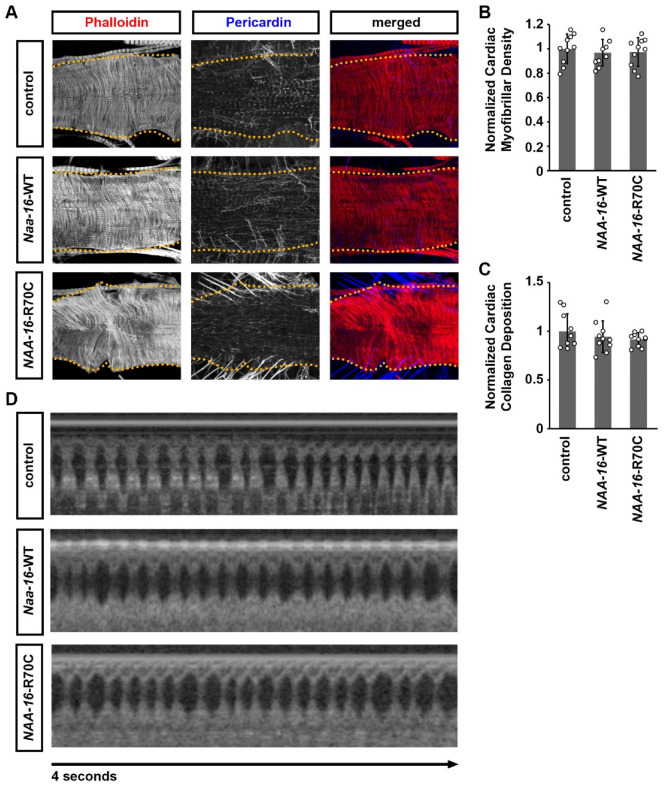
*Drosophila* cardiac phenotypes induced by expression wild-type human *NAA16* and its CHD-associated variant. (**A**) Representative confocal images of adult (4-day-old females) *Drosophila* hearts expressing either wild-type human *NAA16* or its CHD-associated variant R70C. Cardiac actin myofibers were visualized by phalloidin staining (red). Pericardin was detected by immunofluorescence (blue). Dotted lines indicated the boundaries of the heart tube. Control, 4XHand-Gal4+/−. (**B**) Quantitation of adult heart cardiac myofibrillar density relative to control (see image in (**A**)) [mean ± s.d.; *n* = 10 flies (4-day-old females) per genotype; Kruskal–Wallis H-test followed by a Dunn’s test. (**C**) Quantitation of adult heart cardiac collagen (Pericardin) deposition relative to control (see images in (**A**)) [mean ± s.d.; *n* = 10 flies (4-day-old females) per genotype; Kruskal–Wallis H-test followed by a Dunn’s tes. (**D**) Representative OCT images from heartbeat recording of *Drosophila* (4-day-old females) expressing either wild-type human *NAA16* or its CHD-associated variant R70C. Control, 4XHand-Gal4+/−.

**Figure 5 cells-14-01596-f005:**
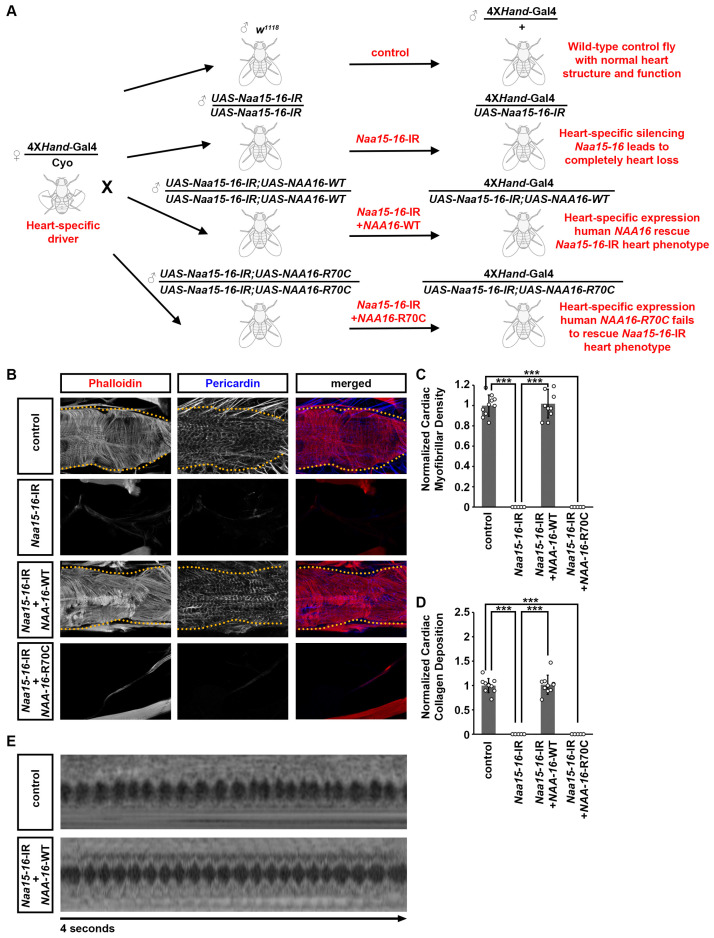
Defects caused by *Drosophila NAA15-16* silencing can be rescued by wild-type human *NAA16*, but not the CHD-associated variant. (**A**) Schematic representation of the “Gene Replacement” strategy used to rescue the heart phenotype caused by heart-specific silencing of *Drosophila NAA15-16*, using human wild-type *NAA16* and the disease-associated variant R70C. (**B**) Representative confocal images of adult (4-day-old females) *Drosophila* hearts UAS-RNAi transgenes targeting *NAA15-16* (4XHand-Gal4), and together with human wildtype *NAA16* or its CHD-associated variant. Cardiac actin myofibers were visualized by phalloidin staining (red). Pericardin was detected by immunofluorescence (blue). Dotted lines indicated the boundaries of the heart tube. Control, 4XHand-Gal4+/−. (**C**) Quantitation of adult heart cardiac myofibrillar density relative to control (see image in (**A**)) [mean ± s.d.; *n* = 10 flies (4-day-old females) per genotype; Kruskal–Wallis H-test followed by a Dunn’s test; statistical significance: *** *p* < 0.001]. (**D**) Quantitation of adult heart cardiac collagen (Pericardin) deposition relative to control (see images in (**A**)) [mean ± s.d.; *n* = 10 flies (4-day-old females) per genotype; Kruskal–Wallis H-test followed by a Dunn’s test; statistical significance: *** *p* < 0.001]. (**E**) Representative OCT images from heartbeat recording of *Drosophila* (4-day-old females) expressing UAS-RNAi transgenes targeting *NAA15-16* (4XHand-Gal4), and together with human wildtype *NAA16* or its CHD-associated variant. Control, 4XHand-Gal4+/−.

**Table 1 cells-14-01596-t001:** NAT complex-related genetic variants identified by PCGC in congenital heart disease. CTD, conotruncal defects; TOF, Tetralogy of Fallot; HTX, Heterotaxy [37].

Patient ID	Cardiac Category	Cardiac Diagnoses	Human Gene	AA Changes	Effect
1-01119	CTD	Atrial septal defect, secundum; Transposition D-loop with ventricular septal defect; Usual coronary arteries in D-loop TGA	NAA16	R70C	Missense
1-01943	TOF	Tetralogy of Fallot	NAA16	L765fs	Frameshift
1-06626	TOF	LSVC to coronary sinus; Tetralogy of Fallot with pulmonary atresia	NAA16	E630fs	Frameshift
1-00455	HTX	Aortic valve position relative to the pulmonary valve, anterior; Atrial inversion; Congenital tricuspid valve abnormality; D-looped ventricles; Dextrocardia; DORV, ventricular defect uncommitted; Heterotaxy; Hypoplastic right ventricle; Hypoplastic tricuspid valve; IDD; Left superior vena cava to right atrium; Pulmonary stenosis, bilateral branch pulmonary artery; Pulmonary stenosis, valvar; Right aortic arch; Right superior vena cava absent; Totally anomalous pulmonary venous return, mixed	NAA15	K336fs	Frameshift
1-00141	TOF	Pulmonary stenosis, valvar; Single left coronary; Tetralogy of Fallot	NAA15	S761X	Nonsense

## Data Availability

The original contributions presented in this study are included in the article/Appendix A. Further inquiries can be directed to the corresponding authors.

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
