# Peer review of "Drosophila Models Reveal NAT Complex Roles in Heart Development and Enable Functional Validation of Congenital Heart Disease Variants"

_cells, 2025, doi:10.3390/cells14201596_

Round 1
Reviewer 1 Report
Comments and Suggestions for Authors
While the study claims to be groundbreaking, it would benefit from a more precise description of how its findings differ from or build on previous work on NAT complexes and cardiac development. Furthermore, the text briefly mentions NAA10 dysfunction but does not elaborate on its mechanistic role or how it was studied, which weakens the claim of "strong in vivo support." Additionally, the text lacks details on experimental controls, statistical validation, and sample sizes, which are essential for assessing the reproducibility and reliability of the results. Furthermore, the term "two-tier platform" is introduced without explanation; clarifying its components would enhance methodological transparency. Also, the claim that the findings "establish" the NatA complex as a critical regulator may be overstated without corroborating evidence from mammalian models. Furthermore, extrapolation to drug discovery, while promising, remains speculative without preliminary screening data. This study presents a compelling case for the NatA complex as a key player in cardiac development, supported by rigorous Drosophila-based experimentation and translational insights. While the results are promising, the claims would benefit from additional mechanistic details, broader validation, and more precise methodological exposition.
Author Response
Comments 1: While the study claims to be groundbreaking, it would benefit from a more precise description of how its findings differ from or build on previous work on NAT complexes and cardiac development.
Responses:In the INTRODUCTION section, we included a paragraph summarizing previous studies that have associated NAT complexes with heart disease and emphasizing how our work differs. Briefly, most prior studies were limited to case reports, and no direct in vivo evidence has demonstrated that NAT complexes are required for heart development. Moreover, a comprehensive investigation to determine which specific NAT complexes contribute to heart development has not yet been undertaken. We have highlighted these distinctions in the revised manuscript.
Comments 2: Furthermore, the text briefly mentions NAA10 dysfunction but does not elaborate on its mechanistic role or how it was studied, which weakens the claim of "strong in vivo support."
Responses:This study focuses on identifying which specific NAT complexes contribute to heart development. The detailed mechanistic roles of these complexes were beyond the scope of the current work. We are, however, planning to conduct a separate study that specifically examines the NatA complex to define its role in heart development, downstream targets, and molecular pathways. We expanded our discussion to include potential pathways through which NatA dysfunction may impair cardiac development, including effects on cytoskeletal integrity, sarcomeric stability, and ECM remodeling. Accordingly, we have adjusted our claim from providing “strong in vivo support” to “in vivo support” and acknowledge that the mechanistic aspects will be addressed in future studies.
Comments 3: Additionally, the text lacks details on experimental controls, statistical validation, and sample sizes, which are essential for assessing the reproducibility and reliability of the results.
Responses:The genotypes of the control flies, along with the sample sizes and ages, are described in the MATERIALS AND METHODS. The statistical analysis subsection of MATERIALS AND METHODS details how the data were analyzed. Additionally, all of this information is also provided in the figure legends.
Comments 4: Furthermore, the term "two-tier platform" is introduced without explanation; clarifying its components would enhance methodological transparency.
Responses:We removed the term “two-tier platform” and instead consistently used “high-throughput platform,” which more accurately reflects the nature of our system.
Comments 5: Also, the claim that the findings "establish" the NatA complex as a critical regulator may be overstated without corroborating evidence from mammalian models.
Responses:We acknowledge this limitation and suggest that further studies in mammalian models are necessary to establish the NatA complex as a critical regulator of heart development in the ABSTRACT. And we have adjusted our claim from “establish” to “suggest” in the ABSTRACT.
Comments 6: Furthermore, extrapolation to drug discovery, while promising, remains speculative without preliminary screening data.
Responses:We acknowledge this point and propose that drug screening can first be performed using the Drosophila platform and subsequently validated in mammalian models.
Comments 7: This study presents a compelling case for the NatA complex as a key player in cardiac development, supported by rigorous Drosophila-based experimentation and translational insights. While the results are promising, the claims would benefit from additional mechanistic details, broader validation, and more precise methodological exposition.
Responses:We thank the reviewer for recognizing the significance of our study. In the revised manuscript, we have addressed the comments in detail and provided additional information as requested.
Reviewer 2 Report
Comments and Suggestions for Authors
In this study, Zhu et al., employs Drosophila as an in vivo model to explore the contribution of NatA complex components to cardiac development and their connection to congenital heart disease (CHD). Targeted knockdown of individual subunits (Naa15-16, vnc, and san) in the heart leads to early lethality, disrupted morphology, fibrosis, and compromised cardiac performance. The edxpression of the human NAA16 gene successfully restored cardiac integrity in flies lacking Naa15-16, whereas the CHD-associated NAA16-R70C variant did not, providing direct functional support for its pathogenic role. Further analyses revealed that Naa15-16 silencing eliminated the adult heart structure, while loss of vnc or san caused severe disorganization of actin filaments, decreased muscle fiber density, and excessive pericardin deposition. Functionally, suppression of vnc or san extended the cardiac cycle without altering chamber diameters. Importantly, overexpression of human NAA16 rescued the structural and functional abnormalities in Naa15-16-deficient hearts, while the R70C variant showed no dominant-negative or gain-of-function properties. The findings identify the NatA complex components as key regulators of heart formation and provide functional evidence connecting human NatA gene variants to CHD.
Concerns:
- Knockdown efficiency – Could you please provide the quantitative knockdown efficiency for each gene compared with the control group? This information is critical to assess the validity of the genetic manipulations.
- Survival curve of Naa15-16-IR – According to the survival curve, approximately 60% of Naa15-16-IR flies survive at day 4, and no flies remain alive by day 10. However, in Figures 2D and 2E, the myofibrillar structure and collagen signal in all Naa15-16-IR samples appear essentially absent, which does not seem representative of the survival data. Could you please clarify this discrepancy?
- Figure 5D recording duration – Please confirm whether the recording duration was 2 seconds or 4 seconds.
- Illustration of NAA16 expression – Adding a schematic figure to show how NAA16 is expressed in Naa15-16-IR individuals would be very helpful for readers to better understand its expression pattern.
- Missing image of Naa15-16-IR – Figure 5D currently lacks the corresponding image for Naa15-16-IR. Please include this for completeness and direct comparison if possible.
Author Response
Comments 1: Knockdown efficiency – Could you please provide the quantitative knockdown efficiency for each gene compared with the control group? This information is critical to assess the validity of the genetic manipulations.
Responses:We conducted additional experiments using quantitative RT-PCR to assess knockdown efficiency relative to the control. Briefly, all RNAi lines showed a 60–70% reduction compared to controls. These results are presented in Supplemental Figure 2, and the manuscript has been updated to include this information.
Comments 2: Survival curve of Naa15-16-IR – According to the survival curve, approximately 60% of Naa15-16-IR flies survive at day 4, and no flies remain alive by day 10. However, in Figures 2D and 2E, the myofibrillar structure and collagen signal in all Naa15-16-IR samples appear essentially absent, which does not seem representative of the survival data. Could you please clarify this discrepancy?
Responses:We used 4-day-old female NAA15-16-IR flies. In these flies, phalloidin staining did not reveal any detectable heart structure. This indicates a complete loss of cardiac myofibrillar organization and cardiac fibrillar density. Furthermore, collagen signals detected with the pericardin antibody were also absent in the hearts of NAA15-16-IR flies. These quantifications correspond to the representative image shown in Figure 2C. We have updated the manuscript to include and highlight these observations.
Comments 3: Figure 5D recording duration – Please confirm whether the recording duration was 2 seconds or 4 seconds.
Responses:We have corrected this error; the value in Figure 5D is 4 seconds.
Comments 4: Illustration of NAA16 expression – Adding a schematic figure to show how NAA16 is expressed in Naa15-16-IR individuals would be very helpful for readers to better understand its expression pattern.
Responses:An additional panel illustrating the “Gene Replacement” strategy has been added to Figure 5A, and both the manuscript and figure legend have been updated accordingly.
Comments 5: Missing image of Naa15-16-IR – Figure 5D currently lacks the corresponding image for Naa15-16-IR. Please include this for completeness and direct comparison if possible.
Responses:Because the Drosophila heart was completely lost due to NAA15-16 silencing, no cardiac signals could be detected using OCT. We have updated the manuscript to reflect this in Fig. 3 and 5.
Reviewer 3 Report
Comments and Suggestions for Authors
The article “The NatA Complex as a Critical Regulator of Cardiac Development: Insights from Drosophila and Congenital Heart Disease Variants” by Jun-yi Zhu and colleagues is an interesting piece of work in which the authors aim to demonstrate that NatA is relevant for Drosophila heart development and to propose how this model can be used to decipher the role of genetic variants in CHD. Despite these efforts, the article presents several gaps that make it unsuitable for publication in its current form. I will outline some of these points, though a deeper critique is required from the authors. Some approaches remain superficial.
-The first result concerning gene expression patterns seems to rely on previous work. If this pattern has already been described, it should be cited rather than reintroduced. If it is new, then more methodological details must be included.
-Regarding the absence of cardiac structures: were changes observed in the aorta/heart domains during embryogenesis? Were alterations in abdominal segment characteristics analyzed?
-Since most of the posterior region of the cardiac tube (corresponding to the heart) is eventually histolyzed, was this process altered in the Naat15-16-IR model? Did it initiate earlier, or is there a lack of initial structures? A deeper description is required.
As anomalies are mainly visualized in adults, were any abnormalities observed in the second instar? What characteristics of the non-eclosed embryonic hearts are relevant?
-What role do van/san play in heart development after metamorphosis? The authors should discuss their results more explicitly in the context of Drosophila heart development.
-Optical coherence tomography is more similar to echocardiography than to electrocardiography. Please correct this misinterpretation.
-The statement that silencing vnc or san did not alter systolic or diastolic diameters but significantly increased the heart period compared to control flies (Fig. 3B) is problematic. Quantification is missing, and some images do not allow assessment of diastolic function.
-Why should Drosophila be considered a model for late cardiogenesis defects, given that the anomalies under investigation depend on segments or structures without direct counterparts in flies? This point requires deeper discussion.
-The International Paediatric and Congenital Cardiac Codes are used but not referenced (e.g., LAT). Proper citation is required.
-More details are needed regarding the variant studied. Ortholog analysis, gnomAD classification, and/or PolyPhen predictions should be included, particularly for p.R70C. Are the same amino acids, functions, or domains conserved between Drosophila and humans?
-Statements such as “Neither construct induced morphological or functional cardiac abnormalities (Fig. 4), indicating that the R70C variant does not function in a gain-of-function or dominant-negative mutation” are limited, as the authors do not provide sufficient evidence.
-How can the authors explain that R70C does not alter heart development but also fails to rescue the phenotype in the Naa15-16 IR model?
-Finally, the title does not fully reflect the presented evidence. Much of the discussion centers on the relevance of Drosophila as a model, rather than specifically on the NatA complex. This should be revised.
Author Response
Comments 1: The first result concerning gene expression patterns seems to rely on previous work. If this pattern has already been described, it should be cited rather than reintroduced. If it is new, then more methodological details must be included.
Responses:The gene expression pattern results were based on a previously published dataset from our group (Huang et al., Single-cell profiling of the developing embryonic heart in Drosophila, 2023, Development), which has been cited in the manuscript. We have highlighted this point in the revised version.
Comments 2: Regarding the absence of cardiac structures: were changes observed in the aorta/heart domains during embryogenesis? Were alterations in abdominal segment characteristics analyzed?
Responses:We thank the reviewer for this valuable suggestion. We agree that identifying heart changes during embryogenesis is important and would greatly enhance our understanding of the roles of NatA complex components in cardiac development. However, the current study was focused on determining which specific NAT complexes contribute to heart development, and detailed analyses of these complexes during embryogenesis were beyond its scope. We are, however, planning to conduct a separate study specifically investigating the NatA complex to define its role in heart development, downstream targets, and molecular pathways. We expanded our discussion to include potential pathways through which NatA dysfunction may impair cardiac development, including effects on cytoskeletal integrity, sarcomeric stability, and ECM remodeling. We have indicated this in the revised manuscript, noting the importance of examining heart changes during embryogenesis in future work.
Comments 3: Since most of the posterior region of the cardiac tube (corresponding to the heart) is eventually histolyzed, was this process altered in the Naat15-16-IR model? Did it initiate earlier, or is there a lack of initial structures? A deeper description is required.
Responses: We agree that it is important to investigate how heart structures disappear in adult Drosophila upon Naa15-16 silencing, as this would further clarify the roles of Naa15-16 in heart development. However, the focus of the present study was to determine which specific NAT complexes contribute to heart development, and detailed analyses of the roles of individual complexes were beyond its scope. We have indicated this in the revised manuscript, highlighting the importance of examining heart changes caused by Naa15-16 silencing in future work.
Comments 4: As anomalies are mainly visualized in adults, were any abnormalities observed in the second instar? What characteristics of the non-eclosed embryonic hearts are relevant?
Responses:We agree that identifying heart changes during development, especially in 2nd instar larval stage is important and would greatly enhance our understanding of the roles of NatA complex components in cardiac development. However, the current study was focused on determining which specific NAT complexes contribute to heart development, and detailed analyses of these complexes during early developmental stages were beyond its scope. We are, however, planning to conduct a separate study specifically investigating the NatA complex to define its role in heart development during different developmental stages, downstream targets, and molecular pathways. We have indicated this in the revised manuscript, noting the importance of examining heart changes during early developmental stages in future work.
Comments 5: What role do van/san play in heart development after metamorphosis? The authors should discuss their results more explicitly in the context of Drosophila heart development.
Responses:To our knowledge, no studies have investigated the roles of vnc and san in Drosophila heart development. Their exact functions at different stages of cardiac development remain unknown. We are planning to conduct a separate study specifically examining the NatA complex, including Naa15-16, san, and vnc, to define its role in heart development across developmental stages, as well as its downstream targets and molecular pathways. This future direction has been noted in the revised manuscript.
Comments 6: Optical coherence tomography is more similar to echocardiography than to electrocardiography. Please correct this misinterpretation.
Responses:We corrected this in the revised manuscript.
Comments 7: The statement that silencing vnc or san did not alter systolic or diastolic diameters but significantly increased the heart period compared to control flies (Fig. 3B) is problematic. Quantification is missing, and some images do not allow assessment of diastolic function.
Responses:We have included images of the heart tube during diastole and systole in Figure 3C. Additional quantifications for vnc and san silenced Drosophila were also provided (Figure 3D and E). Both the manuscript and the figure legend have been updated accordingly.
Comments 8: Why should Drosophila be considered a model for late cardiogenesis defects, given that the anomalies under investigation depend on segments or structures without direct counterparts in flies? This point requires deeper discussion.
Responses:We agree that certain late cardiogenesis anomalies in humans involve chambers or structures that do not have direct morphological counterparts in Drosophila. However, Drosophila is considered a valuable model for studying early cardiogenesis defects because many signaling pathways, transcriptional regulators, and structural proteins involved in vertebrate heart development are highly conserved in flies. Perturbations in these conserved components often produce analogous phenotypes, including defects in myocardial organization, contractility, fibrosis, and arrhythmia.
Importantly, Drosophila allows rapid in vivo testing of gene function and variant pathogenicity in the context of a beating heart. Although the fly heart does not recapitulate chamber-specific features, it provides a genetically tractable and high-throughput system to uncover conserved pathways and validate CHD-associated variants before moving into more complex mammalian models. In the revised manuscript, we have expanded our discussion to emphasize these points and to clearly acknowledge the limitations of the fly system in modeling late cardiogenesis, while highlighting its translational value for identifying conserved genetic mechanisms underlying CHD.
Comments 9: The International Paediatric and Congenital Cardiac Codes are used but not referenced (e.g., LAT). Proper citation is required.
Responses:We have added the full name for this code in the manuscript. LAT was corrected to HTX based on the latest publication of this patient (Jin et al., Contribution of rare inherited and de novo variants in 2,871 congenital heart disease probands, 2017, Nature Genetics). Appropriate references for these codes have been cited.
Comments 10: More details are needed regarding the variant studied. Ortholog analysis, gnomAD classification, and/or PolyPhen predictions should be included, particularly for p.R70C. Are the same amino acids, functions, or domains conserved between Drosophila and humans?
Responses:The variant information was originally reported by Jin et al. in Nature Genetics (“Contribution of rare inherited and de novo variants in 2,871 congenital heart disease probands”). We have updated the manuscript to include all details regarding the R70C variant. Additionally, we compared the amino acid sequences between Drosophila and humans and confirmed that the mutant residue R70 is conserved in both species. This residue is located within the TPR domain, which is also conserved between Drosophila and humans. These updates have been incorporated into the revised manuscript.
Comments 11: Statements such as “Neither construct induced morphological or functional cardiac abnormalities (Fig. 4), indicating that the R70C variant does not function in a gain-of-function or dominant-negative mutation” are limited, as the authors do not provide sufficient evidence.
Responses:Based on our observations, cardiac-specific overexpression of the R70C variant did not produce additional phenotype compared with wild-type NAA16. Therefore, we suggest that this variant is unlikely to represent a gain-of-function or dominant-negative mutation. We have updated the description of the results and revised our statement accordingly.
Comments 12: How can the authors explain that R70C does not alter heart development but also fails to rescue the phenotype in the Naa15-16 IR model?
Responses:The R70C variant did not alter heart development when overexpressed but also failed to rescue the phenotype in the Naa15-16 IR model. This pattern indicates that the variant protein may be neither gain-of-function nor dominant-negative. Instead, our findings suggest that the R70C variant functions as a hypomorphic allele, exhibiting partial loss of activity that is insufficient to sustain normal cardiac development in the absence of endogenous Naa15-16. We have added this explanation to the revised manuscript.
Comments 13: Finally, the title does not fully reflect the presented evidence. Much of the discussion centers on the relevance of Drosophila as a model, rather than specifically on the NatA complex. This should be revised.
Responses:We modified the title to “Drosophila Models Reveal NAT Complex Roles in Heart Development and Enable Functional Validation of Congenital Heart Disease Variants” to better reflect the focus of our study on identifying which specific NAT complexes contribute to heart development and on validating potential CHD-associated variants.
Round 2
Reviewer 3 Report
Comments and Suggestions for Authors
The authors have satisfactorily addressed the raised comments and questions, significantly enhancing the understanding of the manuscript's initial limitations and inquiries.